# Successive Interference Cancellation Based Throughput Optimization for Multi-Hop Wireless Rechargeable Sensor Networks [note 1]

**DOI:** 10.3390/s20020327

**Published:** 2020-01-07

**Authors:** Peng Zhang, Xu Ding, Juan Xu, Jing Wang, Lei Shi

**Affiliations:** 1Institute of Industry and Equipment Technology, HeFei University of Technology, Hefei 230009, China; 2018170902@mail.hfut.edu.cn (P.Z.); dingxu@hfut.edu.cn (X.D.); 2018110996@mail.hfut.edu.cn (J.W.); 2School of Computer and Information, Hefei University of Technology, Hefei 230009, China; shilei@hfut.edu.cn

**Keywords:** network lifetime, successive interference cancellation, multi-hop wireless network, interference management, wireless rechargeable sensor networks

## Abstract

Wireless Sensor Networks are constrained by low channel utilization and limited battery capacity, so they are widely regarded as the mainly performance bottlenecks. In this paper, in order to improve channel utilization and prolong network lifetime, we investigate the cooperation of multi-hop Wireless Rechargeable Sensor Networks (WRSNs) with Successive Interference Cancellation (SIC) technology. In WRSNs, since the flow rate of each node is unknown, the power of the nodes is not constant. However, SIC will not work if the signal power levels at receive node cannot be sorted. To solve this issue, we first construct a minimum energy routing and unify the transmit rate to determine the transmit power. We can also obtain the time scheduling scheme after determining the routing and power. Next, we formulate an optimization problem, with the objective of maximizing the mobile charger’s vacation time over the rechargeable cycle. Finally, we provide a near-optimal solution and prove its feasible performance. Simulation results present that SIC can achieve the better upper boundary on throughput (compared to inference avoidance increasing about 180–450%) and no extra transmit and receive energy consumption in the multi-hop WRSNs.

## 1. Introduction

Wireless Sensor Networks (WSNs) have been attracted increasing attention in the field of theory and application [1]. With the evolution of wireless communication techniques and sensor manufacturing techniques, WSNs are widely used in forest fire alarms [2], disaster self-help [3], environmental monitoring [4], and medical care [5]. Actually, on the one hand, the finite battery capacitance is one of the crucial limitations to the lifetime of WSNs, and it also remains another one of the key bottlenecks to hinder the large-scale deployment of sensor nodes. On the other hand, the interference signal will have an ill influence on the throughput and the channel utilization. Especially, while a large number of sensor nodes are deployed in a limited area, the issue of interference will become serious in WSNs.

For the problem of limited lifetime, the solutions are generally divided into two categories: energy conservation [6,7] and energy harvesting [8]. Energy conservation is an inevitably part of the sensor node’s behavioral model, but it can play a very limited role. In [9], the online and off-line data collection maximization model are proposed for the energy-harvesting sensor networks, but the harvesting energy is calculated by probability rather than a certain value. Therefore, the success of energy harvesting is still limited in practice due to relying intensively on the environment. Further, in the deployment of a WSN, there is also a great concern about the size of the energy-harvesting device, especially whether its size may, at times, exceed that of the sensor device.

For the lifetime of sensor nodes, there is a breakthrough technology on wireless charging technique (WCT) based on magnetic resonant coupling [10]. Kurs et al. showed that energy could be transferred efficiently. Further, they proved both the feasibility and practicality of the WCT. There is a revolutionary influence on WCT on WSNs or other energy-constrained devices, as an infinite lifetime of the WSN can be obtained by periodically recharging. For example, in health-care industry, WCT has already been used to recharge the battery in medical implantable devices [11].

For communication interference, fortunately, Interference Management (IM) [12] gives us an opportunity to reduce the ill effect of interference on open channel communications. IM is a set of many technologies, rather than one single technique, such as Successive Interference Cancellation (SIC) [13], Interference Alignment [14], Interference Coordination [15], etc. The essential idea of IM is to reduce the ill effect of interference signals on the sensor node so that they can transmit or receive more intended data. For instance, R in Figure 1a and $R_{1}$ and $R_{2}$ in Figure 1b are unable to decode the intended signals due to interference. However, if SIC is used, the receive node may implement correctly decoding of all intended signals. When the receive node receives a set of signals from several transmit nodes, it can iteratively decode the received signals if the remaining strongest signal meets the signal-to-interference noise ratio (SINR) threshold (i.e., decodes the strongest signal and views the other signals and the noise as a noise signal). For example, in [16], a heuristic algorithm is proposed for multi-hop wireless network, and Liu et al. showed a bandwidth-aware high-throughput protocol with SIC. In [17], Jiang et al. provided a cross-layer optimization framework with SIC for multi-hop wireless network and presented that the throughput increased about 47%. In [18], Ma et al. employed SIC technology in the Device-to-Device (D2D) cellular network, which is a special wireless network. They studied a framework for the performance in SIC and proposed some general expressions for the successful transmission probabilities in SIC. For the Multiple-Input Multiple-Output (MIMO) degree-of-freedom model, Jalaian et al. [19] eliminated loop interference through SIC technique. We find the SIC technology can make receive nodes receive more intended signals and the wireless networks work more powerfully. In the above article, the transmit power is, however, often a constant, and this transmit power of different transmit nodes is the same. Zhang et al. [20] proposed a quite efficient model, it make SIC well used in multi-hop WRSNs.

In this paper, we analyze a mathematical model for multi-hop WRSNs with SIC technology. The main contributions of this paper are listed as follows:We formulate constraints on the network and the energy consumption model, and we meet the SIC constraints under a different power of the multi-hop WRSNs.Based on this energy consumption model, we construct an optimization model in which the optimization objective is to maximize the vacation time percentage of the mobile charger (MC).We reformulate the original problem into a linear problem with only one quadratic term, and we finally calculate a feasible near-optimal solution to the original problem within the level of error setting in advance.

The rest of this paper is organized as follows: In Section 2, we describe the basic scope of the issue for the multi-hop WRSNs with SIC technique. In Section 3, we propose the mathematical model for a multi-hop WSN with SIC. Section 4 shows the problem formulation and a near-option solution acquired by reformulation. Section 5 presents the numerical results to show the performance for our solution.

## 2. Problem Description

We first introduce the principle of SIC technique in a multi-hop wireless network scenario. Subsequently, based on SIC, we study the energy consumption and supplement model of each sensor node.

### 2.1. Multi-Hop Network Scenario with SIC

There is a fixed base station *B* in the WRSN, as shown in Figure 2, which is the sink for all sensor nodes. Define N as a set of sensor nodes, with $N=\left|N\right|$. Define $E_{max}$ as the battery capacity of each sensor node that all the sensor nodes are fully charged initially. And also denote $E_{min}$ as the threshold of each node keeping alive. The rate of sensing data is $R_{i}$ (in b/s, i∈N) generated by node *i*. Denote $R_{ij}$ as the data flow rate from transmit node *i* to receive node *j* and $R_{iB}$ as the data flow rate from transmit node *i* to the base station *B*. The wireless channel model is the independent Rayleigh fading model. Furthermore, we also suppose that perfect instantaneous channel state information can be obtained. Our future work will relax this supposition to make the network scenarios more practical.

As the number of sensor nodes increases in WSNs, the situation of interference grows severely within the wireless network, which causes a lot of communication conflicts and makes an ill influence on the throughput. Based on signal processing, SIC is a powerful physical technology used in multi-user detection. It allows the receive node to cancel multiple interference signals from different unintended transmit nodes through decoding each signal (receive the intended signal and subtract the unintended signal) iteratively. Based on SIC, the receive node can decode the received superposed signals in order, according to the signal strength. For instance, a receive node decodes the strongest signal from the aggregate received signals and considers all the other signals as noise. If the strongest signal meets the SINR threshold, it can be decoded correctly. Then, the receive node removes it from the set of received superposed signals and rebuilds a new combined signal. And then it repeats the procedure for the second strongest signal, and so forth, until all the intended signals are decoded correctly or the SINR threshold is no longer satisfied at any stage. Therefore, we utilize SIC to eliminate the interference signals in WSNs, so that it increases the throughput.

### 2.2. Sensor Power Supply Model

In the WRSN, we employ a mobile charger (MC) to recharge the battery in each sensor node. During traversal, MC recharges all sensor nodes with the driving speed of *V* (in m/s) from the fixed service station (*S*) to each sensor node in turn and returns to the place of departure *S* to prepare for the next travel (like replacing or rechargeable its battery). Denote τ as one recharge cycle and $\tau_{vac}$ as the vacation time of MC in τ, respectively. Denote $\tau_{i}$ as the rechargeable time of node *i* in τ. After recharging, MC leaves node *i* and travels to the next node. We hypothesize that MC always has enough energy within one single trip.

We aimed to design a rechargeable cycle for MC to obtain more vacation time in the multi-hop WRSNs scenario based on SIC. That is, it is to maximize the vacation ratio in a signal rechargeable cycle ($\frac{\tau_{vac}}{\tau }$).

## 3. The Mathematical Model for SIC Multi-Hop Wireless Networks

In this section, we present the energy consumption model and the time scheduling scheme for a multi-hop WSN with SIC. Energy consumption and the communication model are the sufficient conditions for constructing the complete rechargeable model.

### 3.1. The Mathematical Model for SIC Multi-Hop Wireless Networks

We utilize SIC to improve the network throughput in the physical layer. Simultaneously, we employ the time scheduling scheme in the link layer. Assume one time frame divided equally into *T* time slots. Define $x_{ij}\left[t\right]$ as a binary variable to indicate whether transmit node *i* transmits data to receive node *j* in time slot *t*. $x_{ij}\left[t\right]=1$ if transmit node *i* transmits data to receive node *j* in time slot *t* and 0, otherwise. Similarly, define $x_{iB}\left[t\right]$ as a binary variable. $x_{iB}\left[t\right]=1$ if transmit node *i* transmits data to node the base station *B* in time slot *t* and 0, otherwise.

In time slot *t*, transmit node *i* can transmit to up to one receive node (the sink *B* or relay node *j*), and then we have the following constraint for transmit node:(1)$$\sum_{j\in I_{i}}x_{ij}\left[t\right]+x_{iB}\left[t\right]\leq 1\;\left(i\in N,1\leq t\leq T\right).$$

In the half-duplex model, node *i* cannot transmit nor receive in one time slot. Thus, we have the half-duplex constraint as follows:(2)$$\frac{\sum_{h\in T_{i}}x_{hi}\left[t\right]}{|T_{i}|}+\sum_{j\in T_{i}}x_{ij}\left[t\right]+x_{iB}\left[t\right]\leq 1\;\left(i\in N,1\leq t\leq T\right),$$
where $T_{i}$ is a set of sensor nodes within the transmission range of node *i*.

Based on SIC technique, the receive node can decode multiple intended signals and subtract multiple unintended signals sequentially. During the SIC decoding procedure, receive node *i* (or the base station *B*) attempts to decode the received superposed signals in time slot *t*, and it will preserve the strongest signal (or remove the strongest signal if it is a unintended signal) in turn. Define *g* as the channel loss function, with $g=\alpha \cdot d_{i,j}^{-\lambda }$, where α=1 in general, $d_{i,j}$ is the Euclid distance between transmit node *i* and receive node *j*, and λ is the path loss index. Define $N_{0}$ as the power of the white Gaussian noise. If the SINR of the received superposed signals at receive node *j* is greater than or equal to the certain threshold β, this transmission from transmit node *i* to receive node *j* will be considered as a successful transmission. We then have the SIC constraints of $SINR_{ij}\left[t\right]$ (or $SINR_{iB}\left[t\right]$) at receive node *j* (or the base station *B*) as follows,
(3)$$SINR_{ij}\left[t\right]=\frac{g_{ij}p_{ij}}{\sum_{l\neq i}^{g_{lj}p_{lj}\leq g_{ij}p_{ij}}\left(g_{lj}p_{lj}\sum_{l\in T_{m}}x_{ml}\left[t\right]\right)+N_{0}}\geq \beta \;\left(x_{ij}=1\right),$$
(4)$$SINR_{iB}\left[t\right]=\frac{g_{iB}p_{iB}}{\sum_{l\neq i}^{g_{lB}p_{lB}\leq g_{iB}p_{iB}}\left(g_{lB}p_{lB}\sum_{l\in T_{m}}x_{ml}\left[t\right]\right)+N_{0}}\geq \beta \;\left(x_{iB}=1\right),$$
where $p_{ij}$ is the transmit power of transmit node *i* transmitting data streams to receive node *j*. $\sum_{l\in T_{i}}x_{il}\left[t\right]=1$ if node *i* transmits data in time slot *t*; otherwise, $\sum_{l\in T_{i}}x_{il}\left[t\right]=0$. We hypothesize $x_{ij}=1$ (or $x_{iB}=1$), since the situation, node *i* transmits no data to node *j* (or the base station *B*), does not need to be considered at the time slot.

Within a time frame, the constraint of flow balance at sensor node *i* can be written as follows:(5)$$\sum_{h\in T_{i}}\sum_{k=0}^{l}R_{hi}\cdot x_{hi}\left[t\right]+l\cdot R_{i}=\sum_{j\in T_{i}}\sum_{k=0}^{l}R_{ij}\cdot x_{ij}\left[t\right]+\sum_{k=0}^{l}R_{iB}\cdot x_{iB}\left[t\right]\;\left(i\in N\right).$$

Sensor node energy is mainly consumed by the transmission and reception of data, so we ignore other energy consumptions (such as generating and storing data) for brevity. Define $P_{i}\left[t\right]$ as the power of transmit node *i* in time slot *t*. In this paper, we use the following energy consumption model [21],
(6)$$P_{i}\left[t\right]=c\sum_{h\in N}^{h\neq i}R_{hi}\cdot x_{hi}\left[t\right]+\sum_{j\in N}^{j\neq i}C_{ij}\cdot R_{ij}\cdot x_{ij}\left[t\right]+C_{iB}\cdot R_{iB}\cdot x_{iB}\left[t\right]\;\left(i\in N\right),$$
where *c* is the rate of energy consumption for receiving a unit of data rate from transmit node *i* to receive node *j* (or the base station *B*), and $C_{ij}\;\left(\mathrm{or}\;C_{iB}\right)$ is the rate of energy consumption for transmitting a unit of data rate. Furthermore, $C_{ij}=\chi_{1}+\chi_{2}d_{ij}^{\lambda }$, where $d_{ij}$ represents the Euclid distance between transmit node *i* and receive node *j*, $\chi_{1}$ is a distance-independent constant, and $\chi_{2}$ is a the distance-dependent constant.

Note that $R_{ij}$ and $R_{iB}$ are variables in the optimization problem; in other words, $p_{ij}$ is also a variable and $g_{ij}\cdot p_{ij}$ cannot be sorted in the optimization problem, which means the SINR cannot be determined, and the SIC procedure also cannot be started. Furthermore, since $x_{ij}\left[t\right]$ and $x_{iB}\left[t\right]$ are binary variables, the optimization problem is a mixed integer problem, which is generally an NP-hard problem and cannot be solved straightly.

We show that SIC technique can achieve a better upper boundary on throughput and does not lead to a reduction in the vacation time ratio of MC. Section 6 concludes this paper.

Symbols used in this article and there meanings are listed in Table 1:

### 3.2. Optimization for SIC Multi-Hop Wireless Networks

There are some binary variables and the issue that the optimization variables need to be sorted. That is, the great challenge is that we find a solution for a multi-hop WRSN scenario with SIC. In the subsection, we show the minimum energy routing for calculating the SIC procedure.

Since data rate $R_{ij}$ (or $R_{iB}$) are unknown, the power of node *i* cannot be determined. Lemma 1 can be used to determine the energy consumption of each node merely demanding the amount of data instead of its power.

**Lemma** **1.**
*The total sum of energy consumption of each node is only related to the amount of data which receives and transmits.*


**Proof.** Note that $P_{i}\left[t\right]$ is the power of node *i* at time slot *t*, and during a time frame (i.e., *T* time slots), $\sum_{k=0}^{l}P_{i}\left[t\right]\cdot t_{k}$ is total energy consumption, where $t_{k}$ is the time of one time slot. Therefore, the energy consumption constraint can be represented as follows:
$$\begin{array}{c} \sum_{k=0}^{l}P_{i}\left[t\right]\cdot t_{k}=c\sum_{h\in N}^{h\neq i}\sum_{k=0}^{l}R_{hi}\cdot x_{hi}\left[t\right]t_{k}+\sum_{j\in N}^{j\neq i}\sum_{k=0}^{l}C_{ij}\cdot R_{ij}\cdot x_{ij}\left[t\right]t_{k}+\sum_{k=0}^{l}C_{iB}R_{iB}\cdot x_{iB}\left[t\right]t_{k}, \end{array}$$
where $\sum_{j\in N}^{j\neq i}\sum_{k=0}^{l}R_{ij}\cdot x_{ij}^{k}$ represent the amount of data that transmit node *i* transmits to receive node *j* within one time frame. The three elements on the right side are the sum of energy consumption of the node *i* reception data by intended transmit nodes, the energy consumption of the node *i* for transmitting data to the intended receive node *j*, and the base station *B*, respectively.With determining a route in advance, since the amount of generated data is a constant at each sensor node and it transmits the data streams to the base station *B* straightly or forwards the data streams to the next node, the amount of data streams is a constant.Thus, the sum of energy consumption of each node is only related to the amount of data transmission and reception. In other words, there is no need to care about the rate of transmission and reception.This completes the proof. □

With Lemma 1, we unite all $R_{ij}$ to a constant *R*, then we have the energy consumption constraint as follows:$$P_{i}^{\prime }\left[t\right]=c\sum_{h\in N}^{h\neq i}R\cdot x_{hi}\left[t\right]+\sum_{j\in N}^{j\neq i}C_{ij}\cdot R\cdot x_{ij}\left[t\right]+C_{iB}\cdot R\cdot x_{iB}\left[t\right].$$
with the transmit power of each node, we can calculate the $SINR_{ij}\left[t\right]$ correctly and determine whether $SINR_{ij}\left[t\right]$ meets the criterion threshold. If the data streams can be received and decoded at node *j* correctly (i.e., $SINR_{ij}\left[t\right]\geq \beta$), then we have $x_{ij}\left[t\right]=1$. Similarly, if the data streams can be received and decoded at the sink *B*, then we have $x_{iB}\left[t\right]=1$.

In order to reduce calculation and complexity, we now determine the upper boundary on the number of signals that can be decoded by the sink *B* or relay node *j* in a time slot. Denote $B_{i}$ as the upper boundary on the number of signals.

**Lemma** **2.**
*The number of successful decoded signals at node i is less than or equal to $B_{i}=\min\{1+log_{\beta +1}\left(\frac{P_{i}^{max}}{\beta N_{0}}\right)$, $\left|I_{i}\right|\}$, in a time slot t, where $P_{i}^{max}$ is the strongest power of intended signal in the set of received superposed signal at node i.*


**Proof.** Without loss of generality, suppose that the power levels of the signals from the *k* transmit nodes received at node *i* are in a non-decreasing order as $g_{1i}p_{1i}\leq g_{2i}p_{2i}\leq \cdots \leq g_{ki}p_{ki}$.
$$\begin{array}{cc} g_{1i}p_{1i} & =\beta N_{0}\\ g_{2i}p_{2i} & =\beta \left(N_{0}+g_{1i}p_{1i}\right)=\beta \left(1+\beta \right)N_{0}\\ g_{3i}p_{3i} & =\beta \left(N_{0}+g_{2i}p_{2i}+g_{1i}p_{1i}\right)=\beta {\left(1+\beta \right)}^{2}N_{0}\\ \vdots \\ g_{ki}p_{ki} & =\beta {\left(1+\beta \right)}^{\left(k-1\right)}N_{0}, \end{array}$$Then, we have $k=1+log_{\left(1+\beta \right)}g_{ki}p_{ki}/\beta N_{0}$. In addition, we can find that $g_{ki}p_{ki}\leq P_{i}^{max}$. Since $g_{ki}p_{ki}\leq \beta {\left(1+\beta \right)}^{k-1}N_{0}$, we get the constraint of the number of successful decoded signals as follows,
$$k\leq \beta {\left(1+\beta \right)}^{k-1}N_{0}=B_{i},$$
and define $I_{i}$ as a set of nodes within the interference range of node *i*, which has $\left|I_{i}\right|$ elements. Since the number of received superposed signal cannot exceed the number of adjacent nodes, $B_{i}$ = min{$\left|I_{i}\right|$, $1+log_{\beta +1}\left(\frac{P_{i}^{max}}{\beta N_{0}}\right)$} is the upper boundary on the number of signals which can be decoded by receive node *i*.This completes the proof. □

Based on Lemmas 1 and 2, we utilize the minimum hopping routing to obtain the minimal energy consumption at each node for transmission and reception. Furthermore, by Lemmas 1 and 2 and constraint (Equation 1)–(Equation 10), we can achieve an appropriate time scheduling scheme.

## 4. The Mathematical Model for Recharging Cycle

In this section, we first construct a rechargeable cycle model, then transform the formulation into get a near-optimal solution by change-of-variable technology. The entire rechargeable cycle τ is divided into three parts: the vacation time $\tau_{vac}$, the rechargeable time $\sum \tau_{i}$ and the traversal time $\tau_{tsp}$.

### 4.1. Rechargeable Energy Cycle Construction

We have constructed constraints and lemmas to solve the issue of the dynamic power in the multi-hop WSNs scenario with SIC. In other words, the energy consumption model of each node is complete, and now we start to construct the energy supplement model.

For different rechargeable cycles, node *i* has the same curve of energy consumption. Each node has to comply with the following energy restrictions: In each cycle of τ, (*i*) its remaining battery starts and ends with the same level of energy, and (ii) its remaining battery is always greater than or equal to $E_{min}$.

Define a set of node locations as $\Psi =\left\{\psi_{1},\cdots ,\psi_{n}\right\}$ and define the location of the service station *S* as $\psi_{0}$. Define $d_{\psi_{i},\psi_{\left(i+1\right)}}$ as the distance between two nodes. Denote $P=(\psi_{0},\psi_{1},\cdots ,\psi_{n},\psi_{0})$ as the physical path over a trip cycle, which starts from the service station, passes through each node for recharging it, and finally returns to the service station. In each rechargeable cycle, define $a_{i}$ as MC arrival time at node *i*. Then, we have the constraint as follows:(7)$$a_{\psi_{i}}=k\tau +\sum_{l=0}^{i-1}\frac{d_{\psi_{l},\psi_{l+1}}}{V}+\sum_{l=1}^{i-1}\tau_{l}\;\left(k=0,1,2,\cdots \right).$$

Define $D_{tsp}$ as the physical path distance of the shortest Hamiltonian cycle, thus denoting $\tau_{tsp}=\frac{D_{tsp}}{V}$ as the traveling time over the shortest Hamiltonian cycle. For one singal rechargeable cycle τ, we have the constraint as follows:(8)$$\tau =\tau_{tsp}+\tau_{vac}+\sum_{i\in N}\tau_{i}.$$

During the cycle τ, we analyze the power consumption of node *i* by utilizing the average power ${\bar{P}}_{i}=\frac{\int_{k\tau }^{\left(k+1\right)\tau }P_{i}}{\tau },\;\left(k=0,1,2,\cdots \right)$. The amount of energy consumption of node *i* over τ is equal to the amount of supplement energy over $\tau_{i}$. Therefore, we have the energy balance constraint as follows:(9)$$\tau \cdot {\bar{P}}_{i}=\tau_{i}\cdot u\;\left(i\in N\right)$$

During a rechargeable energy cycle, when MC reaches at node *i* (i.e., $a_{i}$), As shown in Figure 3, there are two rechargeable strategies that can be selected: (i) fully recharge (i.e., MC recharges the battery of node *i* to $E_{max}$ straightly) and (ii) not fully recharge (i.e., MC leaves before fully recharging). These two choices are not different from the results [22]. For the sake of convenience, we select the strategy of fully rechargeable case. When the average power ${\bar{P}}_{i}$ replaces the real-time power $P_{i}\left[t\right]$, it is notable that there are only two approximate slopes of the energy consumption curve over one single rechargeable cycle [$k\tau ,\left(k+1\right)\tau$]: When MC does not recharge node *i*, the energy consumption slope of node *i* is $\left(-{\bar{P}}_{i}\right)$; when MC is recharging node *i*, the energy supplement slope of node *i* is $\left(u-{\bar{P}}_{i}\right)$, where *u* is the power of recharging. The node residual energy cannot exceed the battery capacity $E_{max}$ or be less than the minimum energy threshold $E_{min}$, and the energy constraint can be written as follows:(10)$$E_{min}\leq e_{i}\left(a_{i}\right)\leq e_{i}\left(t\right)\leq e_{i}\left(a_{i}+\tau_{i}\right)\leq E_{max},$$

For a rechargeable energy cycle, $E_{i}=e_{i}\left(2\tau \right)=e_{i}\left(a_{i}+\tau_{i}\right)-\left(2\tau -\left(a_{i}+\tau_{i}\right)\right)\cdot {\bar{P}}_{i}=E_{max}-\left(2\tau -\left(a_{i}+\tau_{i}\right)\right)\cdot {\bar{P}}_{i}$, so $e_{i}\left(a_{i}\right)=E_{i}-\left(a_{i}-\tau \right)\cdot {\bar{P}}_{i}=E_{max}-\left(2\tau -\left(a_{i}+\tau_{i}\right)\right)\cdot {\bar{P}}_{i}-\left(a_{i}-\tau \right)\cdot {\bar{P}}_{i}$. Therefore, the residual energy of the node always needs to be greater than the minimum energy threshold. We have:(11)$$E_{max}-\left(\tau -\tau_{i}\right)\cdot {\bar{P}}_{i}\geq E_{min}\;\left(i\in N\right).$$

The Property 1 in [22] shows that there always exists at least one "bottleneck" node in the WRSN for an optimal solution, and when MC arrives at these nodes and starts to recharge their batteries, the energy level of these nodes is exactly equal to $E_{min}$ (i.e., $E\left(a_{i}\right)=E_{min}$).

**Property** **1.**
*In an optimal solution, there exists at least one node in the network with its battery energy dropping to $E_{min}$ when MC arrives at this node and recharges its battery.*


### 4.2. Mathematical Formulation

We summarize the optimal objective and all the constraints to formulate the optimization problem as $OPT_{raw}$ and show it as follows,
$OPT_{raw}$max$\frac{\tau_{vac}}{\tau }$
s.t.$r_{min}\leq r\left(f\right),\;\left(f\in F\right)$;

Half duplex constraint: (Equation 1), (Equation 2);

Recharging cycle constraint: (Equation 8);

Flow balance constraints: (Equation 5);

Energy constraints: (Equation 6), (Equation 10), (Equation 11);

$\tau ,\tau_{i},\tau_{vac}\leq 0\;\left(i,j\in N,i\neq j\right)$

Variables: $\tau_{i},\tau_{vac},\tau$

Constants: ${\bar{P}}_{i},R,R_{i},c,C_{ij},C_{iB},u,E_{min},E_{max},\tau_{tsp}$

In this problem $OPT_{raw}$, there is a nonlinear objective $\frac{\tau_{vac}}{\tau }$. Note that since the path traveled by MC is independent of the direction of travel along the shortest Hamiltonian cycle, an optimal solution to $OPT_{raw}$ can work in either direction (i.e., clockwise or counterclockwise).

### 4.3. Reformulation

We employ change-of-variable technology to simplify the formulation. For instance, denote $\eta_{vac}=\frac{\tau_{vac}}{\tau }$, replacing the nonlinear objective $\frac{\tau_{vac}}{\tau }$.

For constraint (Equation 8), we divide both sides of the formula by τ, and this constraint can be rewritten as $1=\frac{\tau_{tsp}}{\tau }+\eta_{vac}+\sum_{i\in N}\frac{\tau_{i}}{\tau }$. We, respectively, denote $\eta_{vac}=\frac{\tau_{i}}{\tau }$ and $\eta_{0}=\frac{1}{\tau }$ to replace the nonlinear terms $\frac{1}{\tau }$ and $\frac{\tau_{i}}{\tau }$. Then, constraint (Equation 8) is rewritten as follows:(12)$$\begin{array}{c} \eta_{0}=\frac{1-\sum_{i\in N}\eta_{i}-\tau_{vac}}{\tau_{tsp}}. \end{array}$$

Through the same method, the energy constraints (Equation 10) and (Equation 11) are reformulated as
(13)$$\begin{array}{c} {\bar{P}}_{i}=u\cdot \eta_{i}, \end{array}$$
(14)$$\begin{array}{c} \left(1-\eta_{i}\right)\cdot {\bar{P}}_{i}\leq \left(E_{max}-E_{min}\right)\cdot \eta_{0}\;\left(i\in N\right). \end{array}$$

By constraints (Equation 12) and (Equation 13), we reformulate (Equation 14) as follows.
(15)$$\begin{array}{c} \eta_{vac}\leq 1-\sum_{l\in N}\eta_{l}-\frac{u\cdot \tau_{tsp}}{E_{max}-E_{min}}\cdot \eta_{i}\cdot \left(1-\eta_{i}\right)\;\left(i\in N\right). \end{array}$$

By constraint (Equation 13), constraint (Equation 6) is rewritten as
(16)$$\begin{array}{c} c\sum_{h\in N}^{h\neq i}\sum_{k=0}^{l}\left(R\cdot x_{hi}\left[t\right]\right)+\sum_{j\in N}^{j\neq i}\sum_{k=0}^{l}\left(R\cdot x_{ij}\left[t\right]\right)+C_{iB}\sum_{k=0}^{l}\left(R\cdot x_{iB}\left[t\right]\right)-u\cdot \eta_{i}=0. \end{array}$$

The optimization problem $OPT_{raw}$ is reformulated as $OPT_{S}$, and we show it as follows,
$OPT_{S}$max$\eta_{vac}$
s.t.*Flow balance constraints*: (Equation 5)

*Vaction constraints*: (Equation 15);

*Energy balance constraints*: (Equation 16);

$0\leq \eta_{i},\eta_{vac}\leq 1\;\left(i\in N\right).$
*variables*$\eta_{i},\;\eta_{vac}.$
*constants*$R_{i},\;R,\;c,\;C_{ij},\;C_{iB},\;u,\;E_{max},\;E_{min},\;x_{ij}^{k},\;x_{iB}^{k}\;and\;\tau_{tsp}.$

It is not difficult to find that all the variables in $OPT_{S}$ are transferred from the variables in $OPT_{raw}$.

The constraints (Equation 11) become (Equation 16), which is a linear except, through reformulation (i.e., by change-of-variable technique), but there is still a quadratic term $\eta_{i}^{2}$. In Section 4.4, we will present how to fit the parabola by a useful technology and show a feasible near-optimal solution to $OPT_{raw}$ within a feasible target error.

### 4.4. A Near-Optimal Solution

In this subsection, we replace the parabola (term $\eta_{i}^{2}$) by piecewise straight lines in the reformulation $OPT_{S}$. The approximation transforms the corresponding nonlinear constraints into the linear constraints, so that an off-the-shelf solver, like Gurobi [23], etc., can calculate the solution. Then, we use this solution to determine a feasible solution to the initial problem $OPT_{raw}$. Finally, we prove the feasibility of gap between the solution to $OPT_{S}$ and the optimal solution to $OPT_{raw}$.

It is notable that there is merely one nonlinear term in reformulation $OPT_{S}$ that is the quadratic terms $\eta_{i}^{2}$, and it lies at the interval ([0, 1]), which is a very small interval. Therefore, we fit the nonlinear term $\eta_{i}^{2}$ by a piecewise linear approximation.

The essential idea is replacing the parabola by *m* piecewise linear segments. For the parabola f($\eta_{i}$) = $\eta_{i}^{2}$ ($0\leq \eta_{i}\leq 1$), we connect each point’s ($\frac{s}{m},\frac{s^{2}}{m^{2}}$) to build a piecewise linear approximation. The value of *m* can be determined by Lemma 3 in Section 4.5.

Then, we present a method to mathematically describe ($\eta_{i},\gamma_{i}$) (i.e., the piecewise linear segment). For the sth segment, all points on the piecewise linear curve can be described by the following constraints.
(17)$$\begin{array}{c} \eta_{i}=\lambda_{i,s-1}\cdot \frac{s-1}{m}+\lambda_{i,s}\cdot \frac{s}{m},\;s=0,1,\cdots ,m, \end{array}$$
(18)$$\begin{array}{c} \gamma_{i}=\lambda_{i,s-1}\cdot \frac{{\left(s-1\right)}^{2}}{m^{2}}+\lambda_{i,s}\cdot \frac{s^{2}}{m^{2}},\;s=0,1,\cdots ,m, \end{array}$$
in which $\lambda_{i,s-1}$ and $\lambda_{i,s}$ are two weights, and they satisfy the constraints as follows:(19)$$\begin{array}{c} \lambda_{i,s-1}+\lambda_{i,s}=1, \end{array}$$
(20)$$\begin{array}{c} 0\leq \lambda_{i,s-1},\lambda_{i,s}\leq 1. \end{array}$$

Since the quadratic function $f\left(\eta_{i}\right)=\eta_{i}^{2}$ is convex, the piecewise linear curve ($\frac{s}{m},\frac{s^{2}}{m^{2}}$) lies above the convex curve $f\left(\eta_{i}\right)=\eta_{i}^{2}$, and the end of the approximation curve (i.e., the point $\eta_{i}$ and $\gamma_{i}$) falls on the parabola. For the upper boundary of a feasible error $\gamma_{i}-\eta_{i}^{2}$, Lemma 3 can quantify it [22], if $\gamma_{i}-\eta_{i}^{2}$ is divided into *m* segment [22].

**Lemma** **3.**
*$\gamma_{i}-\eta_{i}^{2}\leq \frac{1}{4m^{2}}\left(i\in N\right)$.*


Although the above formulas can describe $\lambda_{is}$, the expression of is only suitable for a given linear segment (i.e., known the sequence number *s*) and cannot represent a general situation. Now, we present the general mathematical formulas for the entire piecewise linear curve. Denote a binary indicator variable as $l_{is}$ (1≤s≤m). $l_{is}=1$ if $\frac{s-1}{m}\leq \eta_{i}<\frac{s}{m}$; otherwise, $l_{is}=0$. Because $\eta_{i}$ must fall into only one of the *m* segments:(21)$$\sum_{s=1}^{m}l_{is}=1.$$

Based on $l_{is}$, we reformulate $\lambda_{is}$ for the entire piecewise linear curve. We first present the relationship between $\lambda_{is}$ and $l_{is}$ (1≤s≤m). After transforming $OPT_{raw}$ into $OPT_{S}$, we have, at most, two positive values (i.e., $\lambda_{i,s-1}$ and $\lambda_{i,s}$), while the others are $\lambda_{i,j}=0\;\left(j\neq s-1,\;s\right)$. In other words, if $\eta_{i}$ falls only into the first segment at $l_{i1}=1$, we have $\lambda_{i0}>0$; if $\eta_{i}$ falls into the *s*th segment $l_{is}=1$ or $l_{i,s+1}=1$, we have $\lambda_{is}>0$; only if $\eta_{i}$ falls into the last segment $l_{\mathrm{im}}=1$, we have $\lambda_{\mathrm{im}}>0$. These three constraints are formulated as follows:(22)$$\lambda_{i0}\leq l_{i1},$$
(23)$$\lambda_{is}\leq l_{is}+l_{i,s+1},$$
(24)$$\lambda_{im}\leq l_{im}.$$

The constraints (Equation 22)–(Equation 24) indicate that there are at most two positive value $\lambda_{ik}$ for $\eta_{i}$. And if there are the two positive $\lambda_{ik}$, they must be adjacent. Then, for the piecewise linear fitted curve, we rebuild the mathematical expression as follows,
(25)$$\eta_{i}=\sum_{s=0}^{m}\lambda_{\mathrm{is}}\cdot \frac{s}{m},$$
(26)$$\gamma_{i}=\sum_{s=0}^{m}\lambda_{\mathrm{is}}\cdot \frac{s^{2}}{m^{2}},$$
(27)$$\sum_{s=0}^{m}\lambda_{\mathrm{is}}=1.$$

For constraint (Equation 15), substitute $\eta_{i}^{2}$ with $\gamma_{i}$ in $\eta_{vac}\leq 1-\sum_{l\in N}\eta_{l}-\frac{u\cdot \tau_{tsp}}{E_{max}-E_{min}}\cdot \eta_{i}\cdot \left(1-\eta_{i}\right)$; that is,
(28)$$\eta_{vac}=1-\sum_{l\in N}\eta_{l}-\frac{u\cdot \tau_{tsp}}{E_{max}-E_{min}}\cdot \left(\eta_{i}-\gamma_{i}\right)\;\left(i\in N\right)$$

After reformulating these constraints, we have the following linear relaxed formulation called $OPT_{R}$ as follows:
$OPT_{R}$max$\eta_{vac}$
s.t.(Equation 10), (Equation 28), (Equation 21)∼(Equation 27)

$0\leq \eta_{i},\eta_{vac},\gamma_{i}\leq 1\;\left(i,j\in N,i\neq j\right)$

$l_{\mathrm{ik}}\in \left\{0,1\right\}\;\left(i\in N,1\leq s\leq m\right)$

$0\leq \lambda_{ik}\leq 1\;\left(i\in N,0\leq s\leq m\right).$

*Variables*: $\gamma_{i},\eta_{i},\eta_{tsp},\eta_{vac},l_{is},\lambda_{ik}$

*Constants*: $E_{min},E_{max},u,c,C_{ij},C_{iB},x_{ij}^{k},x_{iB}^{k},R$

The off-the-shelf solver, like Gurobi, is able to calculate the solution to formulation $OPT_{R}$ because $OPT_{R}$ has been transformed into a linear formulation from a non-linear formulation.

The two solutions to the linear relaxed formulation $OPT_{R}$ and the reformulation $OPT_{S}$ seem to not be same. In fact, we can derive a feasible near-optimal solution to $OPT_{S}$ through the solution to $OPT_{R}$. Assume the solution to problem $OPT_{R}$ is $\Pi^{\prime }=\left(\eta_{i}^{\prime },\eta_{vac}^{\prime },l_{is}^{\prime },\lambda_{is}^{\prime },\gamma_{i}^{\prime }\right)$. It is notable that ($\eta_{i}^{\prime },\eta_{vac}^{\prime }$) satisfies all constraints in $OPT_{S}$. We attempt to calculate a feasible solution Π = ($\eta_{i},\eta_{vac}$) to $OPT_{S}$; we first assume $\eta_{i}=\eta_{i}^{\prime }$, and then we rewrite $\eta_{vac}$ as follows to meet the constraint (Equation 15) in $OPT_{S}$.
(29)$$\eta_{vac}=\min_{i\in N}\left\{1-\sum_{l\in N}\eta_{l}^{\prime }-\frac{u\cdot \tau_{tsp}}{E_{max}-E_{min}}\cdot \eta_{i}^{\prime }\cdot \left(1-\eta_{i}^{\prime }\right)\right\}.$$

We can calculate a feasible solution to the original problem $OPT_{raw}$ after obtaining a solution to $OPT_{S}$.

### 4.5. Proof of Near-Optimality

We now attempt to quantify the error between the optimal objective to $OPT_{raw}$ and the relaxed objective to $OPT_{R}$. We hope the gap of the error to be controlled by *m* (i.e., we can control the gap by tuning the number of the piecewise linear curve). Though Lemma 4, we reverse the value *m* by a given feasible error ξ (0<ξ≪1).

**Lemma** **4.**
*$\eta_{vac}^{\prime }-\eta_{vac}\leq \frac{u\cdot \tau_{tsp}}{4\left(E_{max}-E_{min}\right)}\cdot \frac{1}{m^{2}}$, where $\eta_{vac}$ is the objective value for the feasible solution *Π* to the original problem $OPT_{raw}$.*


**Proof.** $\eta_{vac}^{\prime }$ is the result for the solution $\Pi^{\prime }$ to $OPT_{R}$, and $\eta_{vac}^{\prime }$ always is greater than or equal to $\eta_{vac}$ (i.e., $\eta_{vac}\leq \eta_{vac}^{\prime }$), since the problem $OPT_{R}$ is a relaxation of the convex reformulation problem $OPT_{S}$. Therefore,
$$\begin{array}{c} \eta_{vac}^{\prime }-\eta_{vac}\leq {\hat{\eta }}_{vac}-\eta_{vac}\\ =[1-\sum_{s\in N}{\hat{\eta }}_{s}-\frac{u\cdot \tau_{tsp}}{E_{max}-E_{min}}\cdot {\hat{\eta }}_{i}\cdot \left(1-{\hat{\eta }}_{i}\right)]-\\ \;[1-\sum_{s\in N}{\hat{\eta }}_{s}-\frac{u\cdot \tau_{tsp}}{E_{max}-E_{min}}\cdot \eta_{max}\cdot \left(1-\eta_{max}\right)]\\ =\frac{u\cdot \tau_{tsp}}{E_{max}-E_{min}}\left(\gamma_{max}-\eta_{max}^{2}\right)\\ \leq \frac{u\cdot \tau_{tsp}}{4\left(E_{max}-E_{min}\right)}\cdot \frac{1}{m^{2}}, \end{array}$$
where the first equality proofed in [22], and the second inequality holds by Lemma 1.This completes the proof. □

Next, we introduce Theorem 1 and present a method to set an appropriate *m* for $\eta_{vac}^{\prime }-\eta_{vac}\leq \xi$ by a given target error ξ.

**Theorem** **1.**
*For a given target error ξ(0<ξ≪1), if $m=\lceil \sqrt{\frac{u\cdot \tau_{tsp}}{4\xi \left(E_{max}-E_{min}\right)}}\rceil$, then we have $\eta_{vac}^{\prime }-\eta_{vac}\leq \xi$.*


**Proof.** Note that the gap is $\eta_{vac}^{\prime }-\eta_{vac}\leq \frac{u\cdot \tau_{tsp}}{4\left(E_{max}-E_{min}\right)}\cdot \frac{1}{m^{2}}$ in Lemma 4. Therefore, set $m=\lceil \sqrt{\frac{u\cdot \tau_{tsp}}{4\xi \left(E_{max}-E_{min}\right)}}\rceil$, and then we have $\eta_{vac}^{\prime }-\eta_{vac}=\xi$:
$$\begin{array}{cc} \eta_{vac}^{\prime }-\eta_{vac} & \leq \frac{u\cdot \tau_{tsp}}{4(E_{max}-E_{min})}\cdot \frac{1}{m^{2}}\\ & \leq \frac{u\cdot \tau_{tsp}}{4(E_{max}-E_{min})}\cdot \frac{4\xi (E_{max}-E_{min})}{u\cdot \tau_{tsp}}\\ & =\xi . \end{array}$$This completes the proof. □

Through Theorem 1, we show the complete procedure on determining the solution to $OPT_{R}$ and present its five steps as follows,
Preset a feasible target error ξ.Set $m=\lceil \sqrt{\frac{u\cdot \tau_{tsp}}{4\xi \left(E_{max}-E_{min}\right)}}\rceil$.Calculate the relaxed linear optimization problem $OPT_{R}$ with *m* linear segment and gain its solution $\Pi^{\prime }=\left(\eta_{vac}^{\prime },\eta_{i}^{\prime },l_{ik}^{\prime },\lambda_{ik}^{\prime },\gamma_{i}^{\prime }\right)$ through the solver *Gurobi*.Construct a feasible solution $\Pi =(\eta_{vac},\eta_{i})$ for the linear fitted problem $OPT_{S}$ by setting $\eta_{vac}=\eta_{vac}^{\prime },\;\eta_{i}=\eta_{i}^{\prime }$ and $\eta_{vac}$ = $\min_{i\in N}${1 - $\sum_{l\in N}\eta_{l}^{\prime }$ - $\frac{u\cdot \tau_{tsp}}{E_{max}-E_{min}}\cdot \eta_{i}^{\prime }\cdot \left(1-\eta_{i}^{\prime }\right)$}Gain a feasible near-optimal solution ($\tau_{vac},\tau ,\tau_{i}$) to the original optimization problem $OPT_{raw}$.

## 5. Simulation

In this section, we present the numerical results to evaluate the performance that our solution can prolong the lifetime of WRSNs to infinity, and SIC technology can improve the upper boundary on throughput by compared with interference avoidance technology.

### 5.1. Simulation Setting

For WRSNs (30, 50, and 80 nodes), we generate the sensor nodes randomly over square areas (0.8 km × 0.8 km, 1 km × 1 km and 1.5 km × 1.5 km). The base station is assumed to be located at (400 m, 400 m) for 0.8 km × 0.8 km, (500 m, 500 m) for 1.0 km × 1.0 km and (750 m, 750 m) for 1.5 km × 1.5 km. The service station is assumed to be located at the origin (0 m, 0 m). Set *V*, the speed of MC, equal to 5 m/s. We assume to randomly generate the data sense rate $R_{i}$ within [1, 20] kb/s and data transmit rate *R* is twice as much as the upper boundary of $R_{i}$ (i.e., R=40 kb/s).

We choose a conventional NiMH battery with $E_{max}=1.08\times 10^{4}$ J and $E_{min}=540$ J $\left(E_{min}=0.05\times E_{max}\right)$. The power consumption coefficients are $\chi_{1}=50$ nJ/b, $\chi_{2}=1.3\times 10^{-3}$ pJ (b·$m^{4}$), c=50 nJ/b, and the path loss index λ=4. For the final numerical results, since we accept the gap between this solution and the optimal solution to $OPT_{raw}$ is no more than 1%, we set the feasible target error ξ=0.01,

### 5.2. Result

For brevity, we first randomly generate a small WSN with 10 sensor nodes over a square area of 0.8 km× 0.8 km, as shown in Figure 4a. The coordinate position and the data generation rate of each node are shown in detail in Table 2. Based on SIC technology, within a time frame (*T* time), the total amount of reception data at the base station increases about 450% compared with that of interference avoidance technology without SIC.

We utilize the solver *Concorde* [24] to get the shortest Hamiltonian cycle for the network with 10 nodes, and the cycle is shown as Figure 5a. For this shortest Hamiltonian cycle, the path distance is $D_{tsp}=2593$ m, and the traveling time of MC is $\tau_{tsp}=518.6$ s. For the presetting target error ξ=0.01, we obtain a small value $m=\lceil \sqrt{\frac{u\cdot \tau_{tsp}}{4\xi \left(E_{max}-E_{min}\right)}}\rceil$ = 4.

Table 3 demonstrates the numerical results of throughput and $\eta_{vac}$ for ten different WRSNs, in which the upper boundary on throughput is obviously increased, while causing no ill effects on objective value $\eta_{vac}$. Since our model and the comparative scheme are using a similar energy consumption model [21] (i.e., formula (Equation 6)), and the amount of data remains the same, the difference of numericals in the experimental results is small. In general, our model is a good combination of SIC technology and WCT, and it has a strong applicability to meet the needs of different scales of WRSNs.

## 6. Conclusions

We investigated the issue of the lifetime of node and the upper boundary of throughput in WRSNs. SIC technology would allow WRSNs to reap the benefits of increasing the upper boundary on throughput, while retaining the rechargeable cycle of WCT without increasing. We researched a optimization problem of practical application with the target of maximizing the ratio of MC’s vacation time over one entire cycle time; however, it is a non-linear problem. Furthermore, we presented and proved a near-optimal solution to the original non-linear optimization problem. The numerical results illustrate SIC technology will increase the upper boundary on throughput immensely and never cause additional energy consumption compared with the scheme of interference avoidance without SIC.

## Figures and Tables

**Figure 1 sensors-20-00327-f001:**
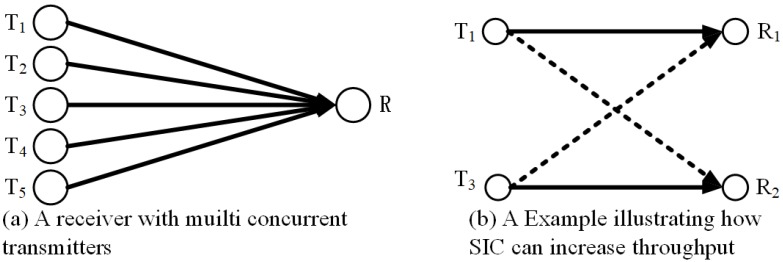
A wireless sensor network with a mobile charger. SIC = Successive Interference Cancellation.

**Figure 2 sensors-20-00327-f002:**
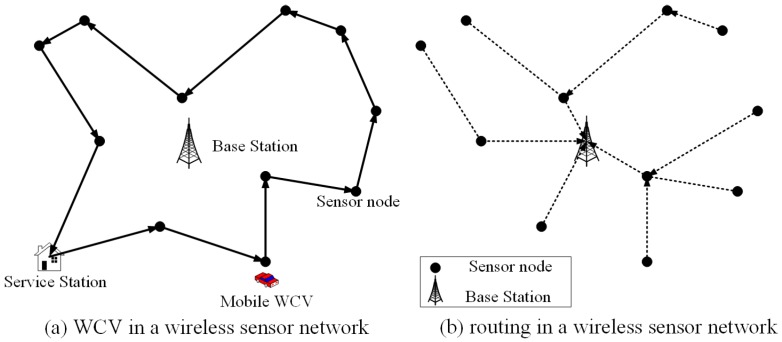
A wireless sensor network with a mobile charger.

**Figure 3 sensors-20-00327-f003:**
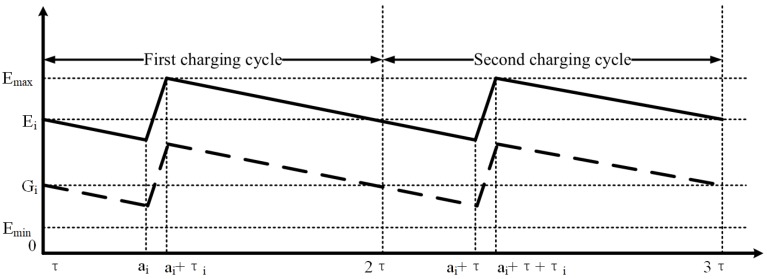
The energy level of a sensor node *i* during two rechargeable cycles (fully re-charged and partially re-charged).

**Figure 4 sensors-20-00327-f004:**
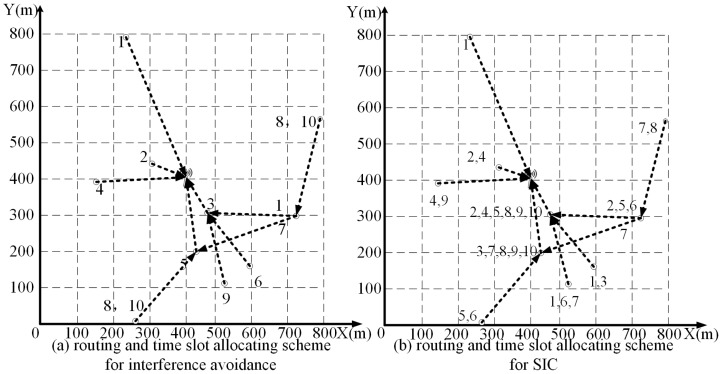
The 10-node sensor network and schemes in the two different techniques.

**Figure 5 sensors-20-00327-f005:**
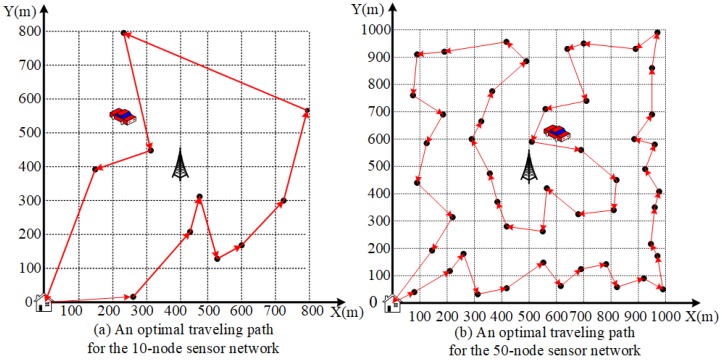
Two optimal traveling paths for the 10-node and 50-node sensor network.

**Table 1 sensors-20-00327-t001:** Notation.

Symbol	Definition
*B*	The base station in the Wireless Rechargeable Sensor Networks (WRSNs)
$B_{j}$	The maximum number of signals for a SIC receive node *j* can decode
*c*	The rate of energy consumption for receiving a unit of data rate
$C_{ij}$	The rate of energy consumption for transmitting a unit of data rate
$g_{ij}$	Channel gain from receive node *i* to transmit node *j*
$I_{i}$	The set of nodes within the interference range of node *i*
N	The set of nodes in the WRSN
$P_{i}$	The power of node *i*
$p_{ij}\left[t\right]$	The transmit power of receive node *i* to transmit node *j* in time slot *t*
$\bar{P_{i}}\left[t\right]$	The average transmission power of node *i* in time slot *t*
$P_{i}^{\prime }$	The transmission power of node *i* with the data rate *R*
$R_{i}$	The data rate generated by node *i*
$R_{ij}$	The flow rate from receive node *i* to transmit node *j*
*R*	The fixed flow rate from receive node *i* to transmit node *j*
$x_{ij}\left[t\right]$	A binary variable indicating weather node *i* is sending data to node *j* in time slot *t*
τ	The time of one single travel cycle for mobile charger (MC)
$\tau_{i}$	The time of recharging node *i*
$\tau_{TSP}$	The time of one single travel through the shortest Hamiltonian cycle
$\tau_{vac}$	The vacation time of MC
$\eta_{vac}$	$\tau_{vac}$ divide by τ (i.e.,$\frac{\tau_{vac}}{\tau }$)
*m*	The number of piecewise linear segments to approximate the parabola
ξ	The presetting target error
$\gamma_{i}$	A substitute for $\tau_{i}$ in piecewise linear approximation
$\lambda_{is}$	The weight about k/m (k=0,1,⋯,m)
$z_{is}$	A binary variable indicating weather $\eta_{i}$ falls within the *k*th segment

**Table 2 sensors-20-00327-t002:** The coordinates (in *m*) and the $R_{i}$ (in kb/s) of each node in the network.

*i*	Coordinates	$R_{i}$	*i*	Coordinates	$R_{i}$	*i*	Coordinates	$R_{i}$	*i*	Coordinates	$R_{i}$
1	(235,635)	4	2	(580,130)	9	3	(311,354)	16	4	(708,240)	5
5	(268, 6)	14	6	(432,160)	17	7	(509,93)	19	8	(775,454)	13
9	(149,312)	13	10	(461,247)	16						

**Table 3 sensors-20-00327-t003:** The numerical results in the four different networks.

The Size of Square	*B*	$\left|N\right|$	$\eta_{\mathrm{vac}}$	The Increasing in Throughput
800×800	(400,400)	10	91.06%	450.00%
800×800	(400,400)	30	89.99%	340.01%
800×800	(400,400)	50	88.01%	334.78%
800×800	(400,400)	80	87.43%	291.43%
1000×1000	(500,500)	30	67.85%	264.78%
1000×1000	(500,500)	50	69.89%	253.34%
1000×1000	(500,500)	80	74.86%	223.26%
1500×1500	(750,750)	80	49.04%	195.62%
1500×1500	(750,750)	100	34.20%	183.37%
1500×1500	(750,750)	150	28.01%	170.83%

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
