# Peer review of "Successive Interference Cancellation Based Throughput Optimization for Multi-Hop Wireless Rechargeable Sensor Networks†"

_sensors, 2020, doi:10.3390/s20020327_

Round 1

Reviewer 1 Report

The paper aims at improving channel utilization and prolong network lifetime through  the cooperation of multi-hop  Wireless Rechargeable Sensor Networks (WRSNs) with Successive Interference Cancellation (SIC). First of all, the authors propose a method to sort the signal power levels at the receiver, then an optimization problem is proposed to maximize the mobile charger’s vacation time over the rechargeable cycle.

The paper addresses an important and actual topic, but I have some concerns/questions:

Which is the channel model and which is its impact on the model and on the performance? This is not clear to me from the paper. Which is the impact of channel state information errors? In multi-hop, which is the impact of the new relay choice? The authors do not consider, in my opinion, recent literature on this aspect which has, instead an important impact on the energy consumption and interference. See, e.g.: Relay selection analysis for an opportunistic two-hop multi-user system in a poisson field of nodes, 2017 or Performance Analysis of Relay Selection in Cooperative NOMA Networks, 2019. The throughput decreases with the square size and, for each square size decreases with N. Is this possible to better highlight these aspects through equations? 

In summary, the paper deserves attention, it deals with an actual topic and proposes an interesting analytical model. However, some aspects should be fixed or better clarified.

Reviewer 2 Report

The paper discusses an interesting topic which is Multi-Hop Wireless Rechargeable Sensor Networks. The proposed approach to throughput optimization uses  Successive Interference Cancellation. The Authors prepared a detailed description and mathematical model of the solution developed during Their research.

The paper is well-written overall. Structure of the paper is correct. the introduction provides good information about current research and problem definition. In the second part of the paper describe theoretical aspects and mathematical model of proposed solution. In the third part of the paper, the simulation results are presented and analyzed. The paper is written clearly and the model is described extensively. However the reviewer recommends that the final version of paper should consist more simulation for different WSN power consumption and recharge schemes to check if the proposed solution is optimal in more set of cases.

The references provided are current and proper for the paper contents.

In reviewer opinion the paper may be published after this minor changes.

Round 2

Reviewer 1 Report

The authors answered to all the reviewer's comments clarifying some aspects and highlighting the importance of their work.